# Information Dark Energy Can Resolve the Hubble Tension and Is Falsifiable by Experiment

**DOI:** 10.3390/e24030385

**Published:** 2022-03-09

**Authors:** Michael Paul Gough

**Affiliations:** Department of Engineering and Design, University of Sussex, Brighton BN1 9QT, UK; m.p.gough@sussex.ac.uk

**Keywords:** Landauer’s principle, dark energy theory, dark energy experiments

## Abstract

We consider the role information energy can play as a source of dark energy. Firstly, we note that if stars and structure had not formed in the universe, elemental bits of information describing the attributes of particles would have exhibited properties similar to the cosmological constant. The Landauer equivalent energy of such elemental bits would be defined in form and value identical to the characteristic energy of the cosmological constant. However, with the formation of stars and structure, stellar heated gas and dust now provide the dominant contribution to information energy with the characteristics of a dynamic, transitional, dark energy. At low redshift, *z* < ~1.35, this dark energy emulates the cosmological constant with a near-constant energy density, *w* = −1.03 ± 0.05, and an energy total similar to the *m*c^2^ of the universe’s ∼10^53^ kg of baryons. At earlier times, *z* > ~1.35, information energy was phantom, differing from the cosmological constant, Λ, with a CPL parameter difference of ∆*w_o_* = −0.03 ± 0.05 and ∆*w_a_* = −0.79 ± 0.08, values sufficient to account for the *H*_0_ tension. Information dark energy agrees with most phenomena as well as Λ, while exhibiting characteristics that resolve many tensions and problems of ΛCDM: the cosmological constant problem; the cosmological coincidence problem; the *H*_0_ tension, and the σ_8_ tension. As this proposed dark energy source is not usually considered, we identify the expected signature in *H(a)* that will enable the role of information dark energy to be falsified by experimental observation.

## 1. Introduction

Many features of the universe are consistent with the standard ΛCDM model. However, as measurements improve in accuracy, a significant difference or tension has been found between the early and late universe Hubble constant, the *H*_0_ values. Planck measurements of the Cosmic Microwave Background, CMB, originating from redshift, z∼1100, provide a ΛCDM model-dependent value for today’s Hubble Constant, *H*_0_, of 67.4 ± 0.5 km s^−1^ Mpc^−1^ [1]. Independent CMB measurements by the Atacama Cosmology Telescope [2] support Planck with a *H*_0_ value of 67.9 ± 1.5 km s^−1^ Mpc^−1^. These values are also consistent with those derived from Baryon Acoustic Oscillations [3].

In contrast to these early universe measurements, *H*_0_ measured in the late universe by a wide variety of techniques yields values closer to 73 km s^−1^ Mpc^−1^. ‘Standard candles’, provided by type 1a supernovae and Cepheid variable stars, offer distance ladders that yield a value of 74.03 ± 1.42 km s^−1^ Mpc^−1^ [4,5]. A recent comprehensive analysis has further confined the value to 73.04 ± 1.04 km s^−1^ Mpc^−1^ [6]. Other methods have been devised to be independent of standard candle/distance ladder techniques. For example, time delay measurements of multiple-imaged quasars due to strong gravitational lensing [7] provide a value of 73.3 + 1.7/−1.8 km s^−1^ Mpc^−1^, differing by 5.3σ from the early universe values. Moreover, the size of edge-on galaxy discs have been determined by the geometry of water maser action occurring in those discs [8], yielding 73.9 ± 3 km s^−1^ Mpc^−1^, a value greater than the early universe value with a 95–99% level of confidence. Another technique using measurements of infrared surface brightness fluctuation distances in galaxies [9] provides a value of 73.3 ± 2.4 km s^−1^ Mpc^−1^, again consistent with other late universe values.

Initially, it was thought that this *H*_0_ tension between the early and late universe might be due to systematic errors [10], but over the last couple of years, the many late universe measurements have become more precise and consistent. The persisting *H*_0_ tension implies a problem or tension with the ΛCDM model, even suggesting new physics beyond ΛCDM [4,11,12]. Possible causes of tension include: a late dynamic dark energy; a universe with non-zero curvature; dark matter interaction; an early dark energy; and additional relativistic particles. Natural theoretical values for the cosmological constant, Λ, differ by several orders of magnitude from the value required to explain accelerating expansion. A likely solution to the tension might then be achieved by replacing Λ with a time-dependent dark energy that has a present energy density compatible with the acceleration.

In this paper, we revisit the role of information energy as the source of such dynamic dark energy [13]. Section 2 reviews the expected present information energy density. Section 3 updates previous work with the latest stellar mass density measurements to determine the equation of state parameters. Section 4 shows that the earlier phantom period can quantitatively account for the observed *H*_0_ tension. Section 5 identifies future measurements necessary to confirm or refute this proposed source of dark energy. The discussion in Section 6 shows that information energy could also resolve the cosmological constant problem and the cosmological coincidence problem.

## 2. Information Energy as Dark Energy

Landauer’s Principle provides an equivalent energy for each bit of information or bit of entropy. Landauer showed that information is physical since the erasure of a bit of information in a system at temperature, *T*, results in the release of a minimum *k_B_ T* ln(2) of energy to the system’s surroundings [14,15]. Landauer’s principle has now been experimentally verified for both classical bits and quantum qubits [16,17,18,19].

A foundational principle has been proposed by Zeilinger [20], whereby the attributes of all particles can be considered at their most basic level as elemental systems with an information content of one bit or qubit. There is a strong similarity between the information energy of such an elemental bit in the universe and the characteristic energy of the cosmological constant [21]. Today, our universe is dominated by dark energy and all matter (baryon + dark), approximately at the ratio of 2:1. If there was no star formation, a representative temperature for matter could be considered to be provided by the temperature, *T_u_*, of a radiation component with the same energy density as all matter:α *T_u_*^4^ = *ρ_tot_* c^2^(1)
where the radiation density constant α = 4σ_SB_/c, σ_SB_ is the Stefan–Boltzmann constant, and *ρ_tot_* is our universe’s total matter density (baryon+dark). σ_SB_ is further defined in terms of fundamental constants:σ_SB_ = π^2^ k_B_^4^/60 ħ^3^ c^2^(2)

Then, we obtain the Landauer equivalent energy of an elemental bit of information in a universe at temperature *T_u_*:k_B_ *T_u_* ln(2) = (15 *ρ_tot_* ħ^3^ c^5^/π^2^)^¼^ ln(2)(3)

This Landauer bit energy is defined identically to the characteristic energy of the cosmological constant. The right-hand side of Equation (3) is identical to Equation 17.14 of [22] for the characteristic energy of the cosmological constant—with the sole addition of ln(2) to convert between entropy units—between natural information units, nats, and bits. Information bit energy might then explain the low milli-eV characteristic energy of Λ, which Peebles [22] considered to be ‘too low to be associated with any relevant particle physics’.

Clearly, the universe is not a simple, single system, and here, we follow a phenomenological approach, taking into account star formation and other information energy contributions. Table 1 lists various astrophysical phenomena, their estimated information bit numbers [23,24], typical temperatures, equivalent information energy total, and that total information energy relative to the universe’s total baryon, *m*c^2^.

It is evident from Table 1 that at present, stellar heated gas and dust make up the dominant information energy contribution, as other sources of information energy are miniscule in comparison. The *N*∼10^86^ bits at typical gas and dust temperatures, *T*∼10^7^, have an equivalent total *N* k_B_ *T* ln(2) energy of ∼10^70^ J, directly comparable to the ∼10^70^ J equivalent *m*c^2^ energy of the universe’s ~10^53^ kg baryons. Stellar heated gas and dust information equivalent energy should then be included alongside the *m*c^2^ equivalence of matter in universe energy accounting. Information energy from stellar heated gas and dust could account for today’s dark energy density using accepted physics, relying solely on the experimentally proven Landauer’s Principle, combined with realistic entropy estimates, and without invoking any new physics at this stage.

The information energy of stellar heated gas and dust has previously been shown [13,25,26,27] to provide a near-constant dark energy density in the late universe, effectively emulating a cosmological constant; at earlier times, this dark energy contribution was phantom. The overall time variation—present constant energy density plus earlier phantom—has been shown to be consistent with the Planck-combined datasets [13]. In the next section, we update that previous work by including more recent stellar mass density measurements.

## 3. Dynamic Information Energy: Time History

In order to include the information energy of stellar heated gas and dust in the accounting of the universe’s energy, we need to describe its variation over time by identifying how total bit number, *N(a)*, and typical temperature, *T(a)*, vary as a function of the universe scale size, *a*, related to redshift, *z*, by *a* = 1/(1 + *z*).

Firstly, we assume that within any sufficiently large volume, the average temperature, *T(a)*, representative of the stellar heated gas and dust, varies in proportion to the fraction, *f(a),* of baryons that have formed stars up to that scale size. We can determine the history of *f(a)* by the plotting in Figure 1, a survey of measured stellar mass densities per co-moving volume [28,29,30,31,32,33,34,35,36,37,38,39,40,41,42,43,44,45,46,47,48,49,50,51,52,53,54,55,56] as a function of scale size, *a*. 

The filled symbols in Figure 1, from source references [28,29,30,31,32,33,34,35,36,37,38,39,40,41,42,43,44], correspond to data compiled for a survey of stellar formation measurements, listed in Table 2 of [57]. A subset of these sources was already used in previous information energy studies [13,26,27], and open symbols, from source references [45,46,47,48,49,50,51,52,53,54,55,56], correspond to measurements used in those previous studies but were not included in the survey of [57].

In Figure 1, there is a significant change around redshift, *z*∼1.35, from a steep gradient in the past to a weaker gradient in recent times. Fitting power laws to data points on either side of *z* = 1.35, we find power laws of *a*^+1.08±0.16^ for *z* < 1.35, and *a*^+3.46±0.23^ for *z* > 1.35. Then, we assume the average stellar heated gas and dust baryon temperature, *T*, proportional to the fraction of baryons in stars, *f(a)*, also varied as *a*^+1.08±0.16^ for *z* < 1.35, and *a*^+3.46±0.23^ for *z* > 1.35. Measured mean galactic electron temperatures over the range 0 < *z* < 1 [58] show a similar temperature time variation as Figure 1, supporting our use of stellar mass densities as a proxy for the gas and dust temperature time variation. 

We consider two possibilities for the time variation of total stellar heated gas and the dust bit number, *N(a)*. In the first case, we assume that *N(a)* simply varies, directly proportional to volume, as *a*^3^. In the second case, we assume that the total bit number of any large co-moving volume is governed by the Holographic Principle [59,60,61] and varies with the volume’s bounding area as *a*^2^. While the Holographic Principle is generally accepted for black holes at the holographic bound, the holographic bound of the universe is ∼10^123^ bits, and the general principle remains only a conjecture for universal application to cases well below the holographic bound [61]. 

We wish to compare the time variation of these information energy models against that of the cosmological constant. The Friedmann equation [62] expresses the Hubble parameter, *H(a)*, in terms of its present value, the Hubble constant, *H*_0_, and dimensionless energy density parameters, Ω, expressed as a fraction of today’s total energy density. Assuming that the curvature term is zero, and that the radiation term has for some time been negligible compared to the other terms, the ΛCDM model is described simply by Equation (4).
**ΛCDM:** (*H(a)/H*_0_)^2^ = Ω_tot_ *a*^−3^ + Ω_Λ_(4)
for present energy fractional contributions, Ω_tot_ from all matter (dark+baryons), and Ω_Λ_ from the cosmological constant.

Total information equivalent energy, given by *N* k_B_ *T* ln(2), is proportional to both *N(a)* and *T(a)*; it is thus proportional to *a*^3^
*f(a)* in the volume model, and proportional to *a*^2^
*f(a)* in the holographic model. These correspond to the information energy density terms Ω_IE_(*f(a)/f(1)*) and Ω_IE_ (*f(a)/f(1)*) *a*^−1^, respectively. Then, if the cosmological constant is negligible and information energy provides the sole source of dark energy, we obtain Equations (5) and (6).
**Information-Volume model:** (*H(a)/H_0_*)^2^ = Ω_tot_ *a*^−3^ + Ω_IE_ (*f(a)/f(1)*)(5)
**Information-Holographic model:** (*H(a)/H_0_*)^2^ = Ω_tot_
*a*^−3^ + Ω_IE_ (*f(a)/f(1)*)*a*^−1^(6)

In Figure 2, we compare the effects of these two models for an information energy source of dark energy against the cosmological constant using the present ΛCDM values, setting Ω_tot_ = 0.32 and Ω_Λ_= Ω_IE_ = 0.68, and applying the power-law fits in Figure 1 for *f(a)*. 

We can see from the upper plot of Figure 2 that the total energy density of the holographic model and that of the cosmological constant nearly coincide, while that for the volume model clearly predicts significantly different total energy densities. The lower plot of Figure 2 emphasizes these differences by plotting the percentage difference in the expected Hubble parameter for the information energy models relative to that of the cosmological constant model. The volume model differs significantly from the cosmological constant, peaking at 7% around *a* = 0.67. Such a difference, at *z* = 0.5, from that expected for a cosmological constant should have easily been observed directly by existing expansion measurements, and for this reason, we hereafter concentrate on the holographic model. The holographic model difference is less than 1% for *a* > 0.4 and for *a* < 0.2, peaking at only 1.8% around *a* = 0.33. 

The time variation of a dark energy density is proportional to *a*^−3(1+*w*)^, where *w* is the equation of the state parameter for that dark energy. Recently, *z* < 1.35, *T(a)* varied as *a*^+1.08±0.16^, *N(a)* as *a*^+2^, hence total stellar heated gas and dust information energy varied as *a*^+3.08±0.16^, providing a near-constant energy density varying only as *a*^+0.08±0.16^, corresponding to the equation of the state parameter, *w* = −1.03 ± 0.05. Then, the information energy of stellar heated gas and dust in the recent period has the characteristics of dark energy, since *f(a)* closely follows the *a*^+1^ gradient that would lead to a near-constant information energy density and emulate a cosmological constant, *w* = −1. Thus, information energy can provide a quantitative account of dark energy, accounting for both the present energy value, ∼10^70^ J, and the recent period of near-constant energy density.

In comparison, during the earlier period, *z* > 1.35, *T* varied as *a*^+3.46±0.23^, total information energy varied as *a*^+^^5.46±0.23^, providing a phantom energy with an energy density increasing as *a*^+2.46±0.23^, corresponding to *w* = −1.82 ± 0.08. The error bars on the equation of the state parameters of both recent and earlier periods are set here by the error bars of the power-law fits in the data of Figure 1. Note that the equation of the state parameter values, *w* = −1.03 ± 0.05 *z* < 1.35 and *w* = −1.82 ± 0.08 *z* > 1.35, are not particularly sensitive to the precise redshift value assumed for the power law break point. For example, choosing break points earlier, *z* = 1.55, or later, *z* = 1.15, the power-law fits to the data of Figure 1 yield very similar values: *w* = −1.06 ± 0.05, *z* < 1.55; *w* = −1.84 ± 0.09, *z* > 1.55; *w* = −1.00 ± 0.06, *z* < 1.15; and *w* = −1.79 ± 0.07, *z* > 1.15.

## 4. Information Energy Can Account for *H*_0_ Tension

Results of experiments to measure the dark energy equation of state, *w*, often assume a simple shape for the *w(a)* timeline, using a minimum number of parameters. Most astrophysical datasets, including Planck data releases [1,63,64,65], have been analyzed to deduce cosmological parameters using the Chevalier, Polarski, Linder, CPL description [66]: *w(a)* = *w*_0_ + (1 − *a*)*w_a_*. This assumes a smooth variation of *w(a)* from *w_o_ + w_a_* at very early times, *a* << 1, through to *w_o_* today (*a* = 1).

The 2013–2018 Planck data releases include several dataset combinations, where Planck data have been combined with other types of measurement and analyzed using the CPL parameters. Although the resultant likelihood regions of *w_o_ − w_a_* space always include the cosmological constant, consistent with ΛCDM, there is a clear overall bias towards an early phantom dark energy (Figure 36 of [63], Figure 28 of [64], Figure 30 of [1]). Most of the likelihood area is located in the space where *w_o_ + w_a_* < −1, the phantom shaded area of Figure 30 in [1].

The information energy equation of the state parameter values, *w* = −1.03 ± 0.05, *z* < 1.35, and *w* = −1.82 ± 0.08, *z* > 1.35, correspond to the CPL parameters, *w*_0_ = −1.03 ± 0.05, *w_a_* = −0.79 ± 0.08, located close to the center of these maximum likelihood regions in *w_o_ − w_a_* space. While the volume model would lead to easily identifiable differences from ΛCDM at low *z*, the holographic model emulates a cosmological constant at low *z,* and for most phenomena, it would be indistinguishable from ΛCDM. The difference between the information energy CPL parameters and those for Λ, *w*_0_ = −1, *w_a_* = 0, is then given by ∆*w*_0_ = −0.03 ± 0.05 and ∆*w_a_* = −0.79 ± 0.08. These parameter differences are significant as they closely match the differences previously considered as the possible means by which a dynamic dark energy could account for the *H*_0_ tension. A dynamic dark energy differing from Λ by ∆*w*_0_ = −0.08 and ∆*w_a_* = −0.8 has been shown to be capable of accounting for much of the ‘H_0_ tension’, from Figure 4 in [4]. Therefore, information dark energy could quantitatively account for the ‘H_0_ tension’.

Note that CPL parameters fit the information *w(a)* values, both today and very early, but information energy exhibits a much sharper transition at *z*∼1.35 than can be faithfully described by CPL. Clearly, the best fit would be provided by the simple sharp transition description: *w* = *w_o_* = −1.03 for *z* < 1.35, and *w* = *w_o_ + w*_a_ = −1.82 for *z* > 1.35. At *z* > 2, dark information energy is negligible, less than 3 percent of total matter energy density; however, it increases rapidly to a near-constant energy density by *z*∼1.35. Such a transitional dark energy, with a sharp change in *w* in the range of 1 < *z* < 2, with a negligible dark energy density at *z* > 2, has previously been shown to be capable of largely accounting for both the ‘H_0_ tension’ and also the ‘σ_8_ tension’ between early and late universe values of the matter fluctuation amplitude [67].

## 5. Information Dark Energy Is Falsifiable by Experiment

The dark energy properties of the information energy identified above might still be just a fortuitous coincidence, and in order to confirm or refute this proposed source of dark energy, we need to predict the value of some future measurement(s).

The main detectable effect of dark energy is the resulting accelerating expansion of the universe. Unfortunately, as information dark energy has closely emulated a cosmological constant in recent times, any differences will be hard to measure. Nevertheless, the clearest verification of information energy’s role as the source of dark energy would be provided by measuring the expected small difference in the Hubble parameter from that of the cosmological constant (Figure 2, lower plot). This difference is a direct result of the earlier phantom period of information energy caused by the steeper stellar mass density gradient at *z* > ∼1.35. This small reduction in *H(a)* is bounded at low redshift by the location of the change of gradient in the Figure 1 measurements, and at higher redshift by the much higher matter energy densities swamping any dark energy contribution.

Power laws were used above primarily to facilitate estimates of the equation of state parameters. Now, we wish to avoid the possibility that the expected difference is an artefact of fitting power laws. Accordingly, in Figure 3, we also apply a sliding average of the Figure 1 measurements (Figure 1, grey line) to generate a data-driven *f(a),* independent of fitted function.

In Figure 3, both methods predict similar values for the signature that would be produced by an information dark energy. The sliding average predicts a maximum difference of −2.2% at *z*∼1.7, while the power-law fits predict a maximum of −1.8% at *z*∼2. There is a clear prediction of a measurable reduction in *H(a)* relative to the cosmological constant over a specific limited redshift range, and hence constitutes a means by which information dark energy can be falsified experimentally.

## 6. Discussion

The present information dark energy value, obtained directly from realistic estimates of bit numbers and temperatures, could account for the accelerating expansion. The late time near-constant energy density follows directly from the measured stellar mass density gradient, providing a *T(a)* variation closely proportional to *a*^+1^, combined with the total *N(a)* proportional to *a*^+2^, assuming the general holographic principle. Strong additional support for an information dark energy is provided by its ability to resolve significant problems or tensions that otherwise remain unexplained and incompatible with the standard ΛCDM model:

### 6.1. H_0_ and σ_8_ Tensions

In Section 4, we saw that an information dark energy can account for much of the ‘H_0_ tension’. The closest CPL parameter description for information energy is identical to the values suggested by Reiss et al. [4] for a dynamic dark energy explanation. A more appropriate description of information energy, accounting for the sharp transition around *z*∼1.35, is identical to the transitional dark energy model [67] that can account for both the H_0_ tension and the σ_8_ tension.

### 6.2. Cosmological Constant Problem

Theoretical estimates for a non-zero value of Λ differ by a massive factor of ∼10^120^ from the actual value required to account for observed accelerating expansion. Despite the lack of any quantitative physical explanation, Λ has been accepted hitherto primarily because of its simplicity and ability to fit the data [68]. Before the expansion of the universe was found to be accelerating, Weinberg [69] considered the most likely value for Λ to be zero. Accounting for all dark energy with information energy would allow Λ to take that preferred zero value, and the information dark energy would effectively resolve the ‘cosmological constant problem’.

### 6.3. Cosmological Coincidence Problem

Star formation needed to have advanced sufficiently for information dark energy to be strong enough to initiate accelerating expansion. Star formation also needed to have advanced sufficiently for there to be a significant likelihood of intelligent beings evolving to observe this acceleration. Therefore, it does not seem to be such a coincidence that we are living when the expansion is accelerating. This effectively removes the ‘why now?’ of the ‘cosmological coincidence problem’ (for example, see [70]).

### 6.4. Cosmic Isotropy

We expect information energy, which is dependent on structure and star formation, to be both temporal and spatially dynamic. Above, we used the universe averaged temporal variation to determine equation of state parameters. Recently, large 5σ significance directional anisotropies have been observed in the value of *H*_0_ [71], calling into question the cosmic isotropy assumption of the cosmological principle. Such directional anisotropies should be expected from the spatial dynamic aspect of an information dark energy located in the stellar heated gas and dust of structures.

### 6.5. Falsifiable

Note that the predicted ∼2% difference in the curve of *H(a)* at *z*∼*2,* as shown in Figure 3, is close to the detection limit of next generation instruments. For example, the ESA Euclid science requirement document [72] states that the aim is to measure *H(a)* down to an accuracy of 1–2% in the range of 0.5 < *z* < 2. Notwithstanding the resolution limits of present instrument designs, this clear prediction will still enable information dark energy to be falsified experimentally in the near future.

Instead of waiting for sufficiently high-resolution measurements of *H(a)* to become available, another method of verifying the role of information dark energy would be to determine whether the observed directional anisotropies in *H*_0_ [71] are related to the distribution of stellar heated gas and dust in the structures of the universe.

### 6.6. Information Dark Energy Compared to Λ and Quintessence

In the discussions above, we have shown that information energy in the late universe closely mimics a cosmological constant. Information energy, IE, can just replace Λ in the ΛCDM model, effectively creating an IECDM model. Then, as the only observable effect of dark energy is via the accelerating expansion, this model should be as consistent with other phenomena as ΛCDM, while also resolving the *H*_0_ and σ_8_ tensions, the cosmological constant and cosmological coincidence problems, and removing the cosmic isotropy assumption of ΛCDM.

Table 2 summarizes and shows that information dark energy compares favourably against the two main dark energy theories: the cosmological constant and scalar fields/quintessence.

Table 2 shows that the equivalent energy of the information/entropy associated with stellar heated gas and dust has many of the characteristics required to be the source of dark energy. In our modified Friedmann Equations (5) and (6), this energy equivalence of information is used in the same way as, and alongside, the *m*c^2^ energy equivalence of matter. We have not needed to identify specific processes nor required information to be destroyed any more than matter needs to be destroyed in order to consider these equivalent energies.

If information dark energy is indeed found to account for the accelerating expansion, then three further aspects should also be considered:

### 6.7. Constant Information Energy Density from Feedback?

The advent of accelerating expansion has been associated with directly causing a general reduction in galaxy merging and a reduction in the growth rate of structure and the rate of star formation [73,74]. This effect is evident in Figure 1 in the clear change of the stellar mass density gradient at *z*∼1.35, from *a*^+3.46±0.23^ to *a*^+1.08±0.16^, coincident with the start of dark energy acceleration effects. Assuming dark energy is information energy, once the information energy density, which increased with star formation, was strong enough to initiate acceleration, the acceleration in turn slowed down star formation, acting as a feedback that directly limited the growth rate of information energy itself. The resulting *a*^+1.08±0.16^ gradient that we observe in Figure 1 is then significant as this range encompasses the specific gradient of *a*^+1^, which should be the natural feedback limited stable value expected from our information energy explanation for dark energy. The constant information energy density at *z* < 1.35, mimicking a cosmological constant, is then a direct result of this feedback limiting. Moreover, in order for feedback to operate in this way, information energy would need to be the major, or sole, source of dark energy.

### 6.8. Can Information Energy Also Emulate Dark Matter Effects?

In this work, we have concentrated on considering the dark energy aspects of information energy. However, another aspect that should be considered is that information energy might contribute to some effects previously attributed to dark matter. We have shown that information energy from stellar heated gas, primarily located around structures, has an energy density at a similar order of magnitude to total matter. Now space-time will be equally distorted by accumulations of matter and equivalent accumulations of energy. Then, information energy will distort space-time, adding to some extent a local attractive force emulating that of gravity from an unseen mass. While on the scale of the universe, total information energy as a dark energy is effectively repulsive thus causing the expansion to accelerate, any extra local distortions to space-time around structures caused by information energy will be effectively attractive and mimic dark matter. Then, by the nature and location of information energy in stellar heated gas and dust, it will be hard to distinguish such effects from those usually attributed to dark matter.

A high correlation has been found in [75,76], showing that dark matter effects in a range of galaxies are fully specified by the location of the baryons. This observation is considered difficult to reconcile with ΛCDM, and the Modified Newtonian Dynamics, MOND, has been suggested as a possible explanation. Equally, this observation might be explained by the information energy of stellar heated gas and dust, contributing effects similar to those produced by dark matter. The strongest dark matter effects in clusters of galaxies are found in the brightest and therefore highest-temperature galaxy [77], again consistent with the highest information energy densities located where stellar heated gas and dust occur at high temperatures and densities.

Clusters of colliding galaxies are often considered to provide some of the strongest evidence for the existence of dark matter. Optical observations show stars pass through the collision largely unhindered, whereas X-ray observations show the galactic gas clouds colliding, slowing down, or even halting. The location of dark matter is then identified from the use of lensing measurements [78,79,80]. A study of the Bullet cluster, and of a further 72 mergers, both major and minor, finds no evidence for dark matter deceleration, with the dark mass remaining closely co-located with the stars and structure. Information energy could equally explain these effects attributed to dark matter, as information energy from stellar heated gas and dust passes, along with the stars, straight through the collision of galaxies. Any contribution of information energy to dark matter effects could be determined by identifying whether the location of the stellar heated gas and dust within galaxies is related to the distribution of dark matter effects observed within those galaxies. New weak-lensing measurements of galaxies [81] promise to measure such effects and distinguish between the various proposed causes.

### 6.9. A Different Future?

The ΛCDM model assumes that universe expansion will continue accelerating forever in this dark energy-dominated epoch. A dark energy provided by the information energy of stellar heated gas and dust suggests a different future for the universe. The fraction of baryons in stars must stop increasing at some time, since *f(a)* < 1 by definition. Eventually, *f(a)* will decrease as more stars die out than new ones are formed. It is estimated that the future maximum star formation might be as little as only 5% higher than today [82]. At some point, the information dark energy density will fall, and the expansion of the universe will cease accelerating and revert back to deceleration.

## 7. Summary

The approach employed in this work has emphasized the two preferred requirements of cosmology [68]: simplicity (wielding Occam’s razor), and naturalness (relying on mostly proven physics, with a strong dependence on empirical data). The information energy of stellar heated gas and dust could provide a dynamic dark energy that overcomes several of the problems and tensions of ΛCDM. It is therefore important to consider performing the falsification measurements suggested above so that such a simple concept can be confirmed, or refuted.

## Figures and Tables

**Figure 1 entropy-24-00385-f001:**
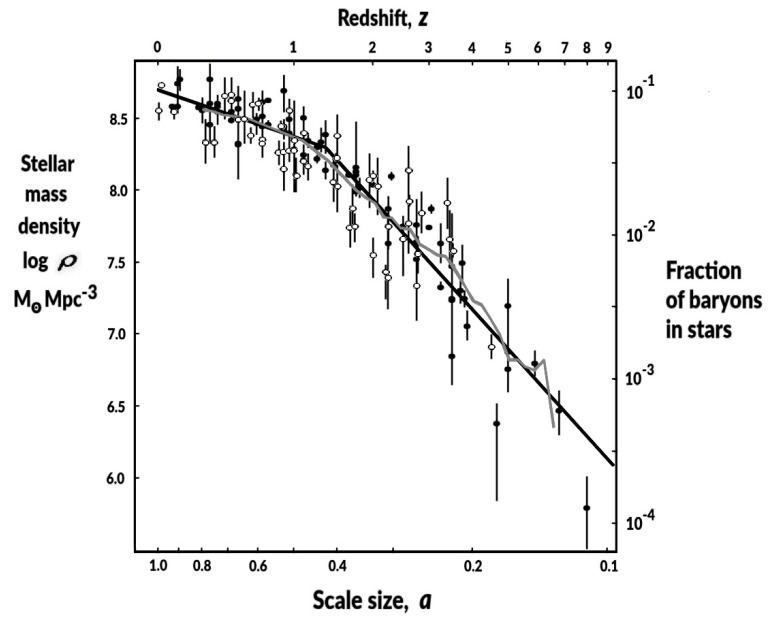
Review of stellar mass density measurements for co-moving volumes as a function of universe scale size, *a*. Straight black lines are power-law fits: *a*^+1.08±0.16^, for *z* < 1.35; and *a*^+3.46±0.23^, for *z* > 1.35. Grey line is the sliding average. Source references: *Filled circles*: [28,29,30,31,32,33,34,35,36,37,38,39,40,41,42,43,44]; *Open circles*: [45,46,47,48,49,50,51,52,53,54,55,56], see text.

**Figure 2 entropy-24-00385-f002:**
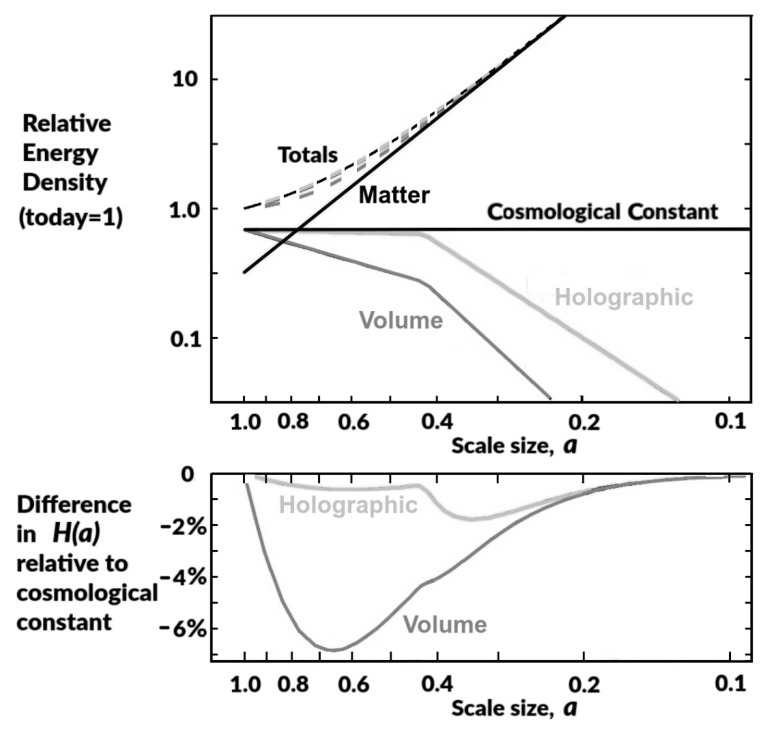
***Upper Plot*.** Energy densities relative to total today (=1.0) for the cosmological constant, information energy, and all matter; the plot includes totals (dashed lines) for all matter+cosmological constant, and all matter+information energy, for the volume and holographic models. ***Lower Plot***. Difference in the Hubble parameter, *H(a),* to be expected from an information energy source of dark energy relative to that resulting from a cosmological constant. Both plots assume the power-law fits in Figure 1 data.

**Figure 3 entropy-24-00385-f003:**
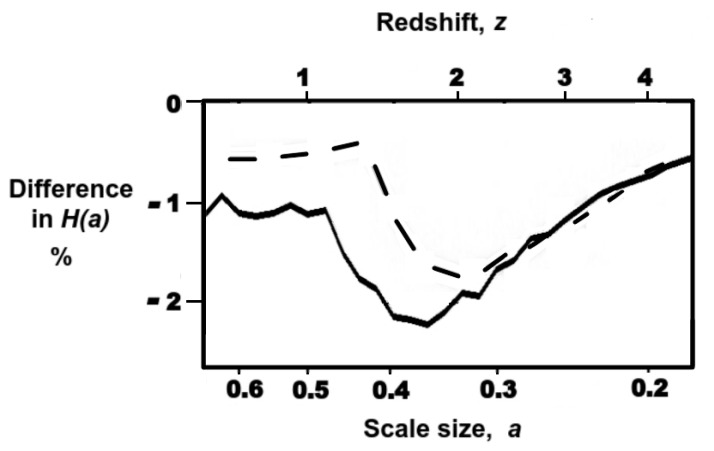
Difference in *H(a)* expected between information energy and the cosmological constant sources of dark energy. Solid black line derived from the sliding average of the data in Figure 1, and dashed line from the power-law fits used previously.

**Table 1 entropy-24-00385-t001:** Present information, temperature, and information energy contributions.

	Information, *N,* Bits	Typical Temperature *T,* °K	Information Energy *N* k_B_ *T* ln2, Joules	Information Energy/ Universe Baryon *m*c^2^
Stellar heated gas and dust	~10^86^	~10^7^	~10^70^	~1
10^22^ stars	10^79^–10^81^	~10^7^	10^63^–10^65^	10^−7^–10^−5^
Stellar black holes	10^97^–6 × 10^97^	~10^−7^	10^67^–6 × 10^67^	10^−3^–6 × 10^−3^
Super massive black holes	10^102^–3 × 10^104^	~10^−14^	10^65^–3 × 10^67^	10^−5^–3 × 10^−3^
Cold dark matter	~2 × 10^88^	<10^2^	<10^67^	<10^−3^
CMB photons	10^88^–2 × 10^89^	2.7	3 × 10^65^–6 × 10^66^	3 × 10^−5^–6 × 10^−4^
Relic neutrinos	10^88^–5 × 10^89^	2	2 × 10^65^–10^67^	2 × 10^−5^–10^−3^

**Table 2 entropy-24-00385-t002:** Comparison of information energy to two main dark energy theories.

Dark Energy Property Required to Fit Observations	Cosmological Constant	Scalar Fields/ Quintessence	Information Energy
Account quantitatively for present dark energy value	Not by orders of magnitude	Only by much fine tuning	Yes, directly ∼10^70^ J
Resolve ‘Cosmological constant problem’	No	Only by much fine tuning	Yes Λ→0
Late universe near-constant dark energy density, *w*~−1	Yes, by definition, *w* = −1	Not specific −1 < *w* < +1	Yes *w* = −1.03 ± 0.05
Consistent with *Planck w*_0_ *− w_a_* parameter likelihood region	Yes	Not specific	Yes
Resolve late vs. early universe ‘*H*_0_ and σ_8_ tensions’	No	No	Yes, quantitatively
Resolve ‘Cosmic coincidence problem’	No	Only by much fine tuning	Yes, naturally
Account for *H*_0_ anisotropies that conflict with ‘cosmic isotropy’	No	No	Yes, expected
Experimentally Falsifiable?	No	No	Yes, Predicted *H(a)*

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
