# Peer review of "Information Dark Energy Can Resolve the Hubble Tension and Is Falsifiable by Experiment"

_entropy, 2022, doi:10.3390/e24030385_

Round 1

Reviewer 1 Report

In the present study, the author has mainly focused his attention on discussing whether the information energy can be taken into account as a source of dark energy. I read the manuscript with interest and care, as it deals with an interesting topic. On the other hand, i have also scanned the entire document via the iThenticate for the ethical status of the submission and conluded that there is no any ethical issue. I think that the paper is well written and presents noteworthy results. I recommend publication of the submission in the Entopy journal.

Best regards.

Author Response

No changes required

Reviewer 2 Report

The paper is clearly written and worth publication.
A few formal remarks that require formal clarifications are the following:

  • Already in the abstract appears the acronym CPL, which returns in various parts of the paper. It is not obvious for the general reader that is stands for  Chevalier, Polarski, Linder. It would be appropriate to clarify it on the first appearance.
  • Similarly shorthand notations such as "nat" should be explicited, at least the first time they appear (line 94, page 3): natural unit of information.
  • Line 56 (page 2) "dependant" should rather be "dependent".
  • Line 72 (page 2) probably "principal" is intended to be "principle".
  • the use of σ for the Stefan-Boltzmann constant conflicts with the successive use for the standard deviation and related quantities.
  • In table 1 a component like Relic gravitons, cannot be considered ad being standard, even though maybe it does not influence the conclusions of the paper.
  • In the caption of fig. 1 and in the text immediately below the figure "open circles" and "open symbols" are mentioned, but they cannot be identified in the figure.
  • In general, apart from having an interesting quantitative correspondence with the dark energy density, it is not clear by which physical mechanisms the energy information should produce an effect like the expansion of the visible universe. The information energy in fact is required and retrieved from the environment to build information and is released to the environment when the information is erased. How this mimics dark energy is not obvious. A comment should in order at least in the discussion.

Author Response

All of the suggested corrections have been made, most identified with red text:

CPL, nats, dependent, principle, using σSB for the Stefan-Boltzmann constant ,

and removed relic gravitons row from Table 1,

I have replaced Fig. 1 with the correct figure that distinguishes open/closed circle data sources,

A comment on the difficulty of understanding information energy as dark energy is included just after Table 2 of section 6.6.

“Table 2 shows that the equivalent energy of the information/entropy associated with stellar heated gas and dust has many of the characteristics required for the source of dark energy. In our modified Friedmann equations, 3.2 and 3.3, this energy equivalence of information is used in the same way as, and alongside, the mc2 energy equivalence of matter. We have not identified specific processes, nor required information to be destroyed any more than matter to be destroyed in order to consider these equivalent energies.”

Reviewer 3 Report

Referee report on the paper

Information Dark Energy can resolve Hubble tension and is falsifiable by experiment

by Paul Gough

This is an interesting work about using information energy to solve the ‘Hubble tension’. The present paper is a follow up and update of his earlier contributions (ref 13 and 25-27) to this topic published in Entropy in the years 2008-2014. In my opinion it can be published in Entropy after a minor revision.

I have a suggestion which I think the author should consider.

In lines 76-78 the author writes: “If there was no star formation, a typical representative  temperature for our matter dominated universe could be considered to be provided by the temperature of a radiation dominated universe with the same energy density as our universe:”

I would like the author either to change “our matter dominated universe” to “our dark energy dominated universe” or explain why he writes “our matter dominated universe”.

Author Response

I agree with the referee that the previous lines 76-78 needed to be changed.

I have changed these lines to:

"Today our universe is dominated by dark energy and matter, approximately in the ratio 2:1. If there was no star formation, a representative temperature for the matter could be considered to be provided by the temperature of a radiation component with the same energy density as the matter:"

I have highlighted this change with red text.